# Isopropoxy Benzene Guanidine Ameliorates *Streptococcus suis* Infection In Vivo and In Vitro

**DOI:** 10.3390/ijms24087354

**Published:** 2023-04-16

**Authors:** Ning Han, Jie Li, Feifei Zhao, Yangyang Li, Jun Wang, Xiaolan Dai, Dongping Zeng, Wenguang Xiong, Zhenling Zeng

**Affiliations:** 1Guangdong Provincial Key Laboratory of Veterinary Pharmaceutics Development and Safety Evaluation, College of Veterinary Medicine, South China Agricultural University, Guangzhou 510642, China; hn18879897170@163.com (N.H.);; 2National Laboratory of Safety Evaluation (Environmental Assessment) of Veterinary Drugs, College of Veterinary Medicine, South China Agricultural University, Guangzhou 510642, China; 3National Risk Assessment Laboratory for Antimicrobial Resistance of Animal Original Bacteria, College of Veterinary Medicine, South China Agricultural University, Guangzhou 510642, China

**Keywords:** *Streptococcus suis*, isopropoxy benzene guanidine, antibacterial, anti-hemolysis, suilysin

## Abstract

*Streptococcus suis*, an encapsulated zoonotic pathogen, has been reported to cause a variety of infectious diseases, such as meningitis and streptococcal-toxic-shock-like syndrome. Increasing antimicrobial resistance has triggered the need for new treatments. In the present study, we found that isopropoxy benzene guanidine (IBG) significantly attenuated the effects caused by *S. suis* infection, in vivo and in vitro, by killing *S. suis* and reducing *S. suis* pathogenicity. Further studies showed that IBG disrupted the integrity of *S. suis* cell membranes and increased the permeability of *S. suis* cell membranes, leading to an imbalance in proton motive force and the accumulation of intracellular ATP. Meanwhile, IBG antagonized the hemolysis activity of suilysin and decreased the expression of *Sly* gene. In vivo, IBG improved the viability of *S. suis* SS3-infected mice by reducing tissue bacterial load. In conclusion, IBG is a promising compound for the treatment of *S. suis* infections, given its antibacterial and anti-hemolysis activity.

## 1. Introduction

*Streptococcus suis* is a common zoonotic pathogen, and its infection is associated with mortality in humans and animals [1]. *S. suis* can cause a variety of diseases in humans and pigs, including pneumonia, meningitis, endocarditis, and streptococcal-toxic-shock-like syndrome, resulting in morbidity and mortality [2,3]. Among the existing *S. suis* serotypes, type 2 is recognized as the most virulent and impactful serotype [4]. There are currently 35 capsular antigen-based serotypes of *S. suis*, of which serotypes 1, 2, and 14 are the most clinically relevant due to their virulence factors [5,6,7]. The major virulence determinants of *S. suis* identified to date are hemolysin (suilysin), muramidase-released protein, and extracellular protein factor [8]. A novel surface protein Fhb (factor H-binding protein), a potential new virulence factor, was identified in Streptococcus serotype two and serotypes one, seven, and nine, contributing to *S. suis* resistance to phagocytosis and virulence [9]. Multiple locus sequence typing (MLST) was used to study the population structure and genetic diversity of *S. suis* [10]. ST25, ST28, ST29, ST94, ST108, ST117, ST225, ST373, ST961, ST977, ST21, and ST31 were found to be associated with pathogenic pathotypes [11]. Pigs are important food animals, and *S. suis* can be transmitted to humans from sick pigs through human skin wounds and nasal mucosa [12]. At present, the human infection with *S. suis* has been widely reported worldwide [13]. With the rapid development of resistance, humans have encountered great challenges in treating infections caused by *S. suis* [14]. Therefore, there is an urgent need for the discovery and development of drugs against *S. suis* infection.

At present, the discovery of antibacterial drugs is mainly based on the following aspects: firstly, the screening and extraction of active compounds from natural products; secondly, the generation of compound libraries by chemical synthesis and the screening of effective compounds; thirdly, the exploration of unknown antibacterial targets and the performance of target screening [15,16,17]. For example, daptomycin, originally discovered in *Streptomyces roserosi*, is a cyclic lipopeptide antibiotic currently used to treat serious infections caused by Gram-positive bacteria, including methicillin-resistant *Staphylococcus aureus* (MRSA), vancomycin-resistant *Staphylococcus aureus*, vancomycin-resistant *Enterococcus*, etc. [18,19,20]. At the same time, drug repurposing has also become an innovative way of drug development [21]. To date, many efforts have been made to screen FDA-approved drugs for antimicrobial and antiviral activity [22,23]. The antiplatelet drug ticlopidine was found to have anti-MASR effects by inhibiting wall teichoic acid biosynthesis in combination with β-lactam antibiotics [24].

In the history of antibiotic discovery, it is easy to see that natural products are representative of the substances used to discover antibiotics. However, natural products have some unavoidable drawbacks; firstly, their wide distribution in nature and the high cost of obtaining them; and secondly, their complex chemical structures, which are difficult to obtain via chemical synthesis. Compared to the natural product library, synthetic compound libraries have a much wider scope, both in terms of quantity and chemical structure variability, and we can modify their structures by conformational relationships to obtain more active compounds [25]. Guanidine functional groups are positively charged and can lead to the loss of biological function of phospholipids or disruption of membranes by binding to negatively charged cytoplasmic membranes or cell wall components, ultimately leading to cell death [26]. The guanidine group is one of the preferred functional groups in the design and development of antibacterial drugs. In our previous study, we screened isopropoxy benzene guanidine (IBG) from a series of guanidinium-based compounds with anti-Gram-positive bacterial activity, such as *Staphylococcus aureus* (0.125–4 µg/mL) and *Enterococcus* (2–16 μg/mL), by disrupting cell membranes [27,28]. This compound shows good drug properties and the possibility of being a leading compound in terms of antibacterial activity and safety. The aim of this study was to further investigate the antibacterial activity and antibacterial mechanism of action of IBG against *S. suis* through serial passages, assessment of the effect on the hemolytic activity of *S. suis* and a mouse model of intraperitoneal infection.

## 2. Result

### 2.1. IBG Is a Potential Antimicrobial Agent

IBG showed strong antibacterial activity against all *S. suis* isolates tested, with MICs ranging from 0.25–8 μg/mL. The MIC of IBG against *S. suis* ATCC 43765 was 8 μg/mL. The activity of IBG against *S. suis* was not potentially related to its multi-locus sequence type (MLST; Table 1). In addition, serum concentration has a greater effect on the bactericidal activity of IBG. In the absence of serum, the MIC of IBG was 2 µg/mL, whereas, in the presence of 10% serum, the MIC increased 8-fold to 16 µg/mL. In resistance studies, IBG did not show strongly induced resistance, and its MIC only increased only fourfold within 28 days and then recovered (Figure 1B). In contrast, the MIC of CIP increased 1024-fold within 28 days. The time-kill kinetics assay showed that IBG had a significant bactericidal effect that was concentration-dependent (Figure 1C). IBG was able to kill most cells at a concentration of 10 × MIC (80 μg/mL) for 4 h. The bactericidal activity of IBG was weaker at 1× and 4 × MIC. At 1 × MIC, there was a weak bactericidal effect within two hours of exposure, after which bacterial growth resumed. Toxicity is one of the key factors limiting the clinical use of drugs [29]. The results of the safety assessment showed that IBG had a very high cut-off range (Figure 1C). It had no apparent hemolytic effect at low concentrations. At the concentration of 10 × MIC, the hemolysis rate was less than 20%. In general, IBG is a promising antibacterial compound.

### 2.2. IBG Disrupts S. suis Cell Membrane

Membrane damage was reflected by an increase in the fluorescence signal of *S. suis* cells stained with PI. As shown in Figure 2E, the fluorescence signal was greatly enhanced in the presence of IBG in a dose-dependent manner compared to the IBG-untreated group. The effect of IBG on PMF was then measured. ΔΨm and ΔpH are two important factors of PMF [30]. The fluorescent probe DiSC_3_(5) was used to detect changes in ΔΨm. The results showed that IBG significantly (*p* < 0.01) decreased the fluorescence value of DiSC_3_(5), especially at 80 μg/mL, indicating that IBG disrupted the ΔΨm of the cell membrane (Figure 2A). ΔpH was then detected using the fluorescent probe BCECF-AM. IBG significantly dissipated ΔpH in a dose-dependent manner (Figure 2B). Since PMF disruption can affect cellular ATP [31], the levels of intracellular and extracellular ATP were measured. The results showed that, correspondingly, IBG significantly decreased the level of intracellular ATP in a dose-dependent manner (Figure 2C). In contrast, a dose-dependent increase in extracellular ATP was observed (Figure 2C). As the generation of ROS is thought to be a typical mechanism of bactericidal antibiotics [32], we next investigated the effect of IBG on ROS levels. The results showed that, as with classical antibiotics, IBG significantly (*p* < 0.001) increased the level of ROS in a dose-dependent manner after 60 min of treatment (Figure 2D).

### 2.3. Significant Anti-Hemolysis Activity of IBG against S. suis

The effect of IBG against suilysin was assessed by hemolysis assays. As shown in Figure 3A, *S. suis* ATCC 43765 culture supernatant had strong hemolytic activity, and approximately 41.36% of sheep erythrocytes were lysed. However, in the presence of IBG, the hemolytic activity of the supernatant was significantly (*p* < 0.0001) decreased in a dose-dependent manner. When the final concentration of IBG added was 80 μg/mL, the hemolytic capacity of the supernatant was only 5.10%. We then examined the effect of IBG on *S. suis* ATCC 43765 *Sly* gene expression. The results showed that IBG could significantly (*p* < 0.01) reduce the expression of the *Sly* gene in a dose-dependent manner compared to the control group (Figure 3B). In addition, the expression of the *Sly* gene results in the production of the Sly protein, a cholesterol-dependent toxin that destroys cells by creating microscopic pores in the host membrane containing cholesterol [33]. Subsequently, molecular docking was performed to analyze the binding between IBG and the Sly protein. The results showed that IBG had a good affinity for the Sly protein with a LibDockScore of 82.4514. Meanwhile, molecular docking further revealed the potential interaction of IBG with the Sly protein. Potentially critical active residues were observed to be involved in the binding sites of IBG, such as VAL232, TYP264, PRO235, LEU124, LYS240, GLY100, THR125, ASP233, GLU234, PRO145, and ASP123 (Figure 3C,D).

### 2.4. In Vivo Efficacy

The in vivo therapeutic efficacy of IBG was evaluated using a mouse model of *S. suis* SS3 infection. The protective effect of IBG on infected mice was assessed by survival rate. Compared to untreated infected mice, 80 mg/kg of IBG increased the survival rate of infected mice by 66%. In testing the degree of infection for different tissues of mice from *S. suis* SS3, SS3 caused severe pathological damage in mice, and a large number of bacterial colonies were found in lung, liver, kidney, and spleen tissues. By treating infected mice with IBG, we found that IBG significantly alleviated the pathological damage caused by SS3. The bacterial load of IBG-treated mouse tissues was significantly (*p* < 0.001) reduced, particularly in the lung and spleen (Figure 4). The survival rate of IBG-treated mice was higher than that of untreated mice, while the survival rate of mice increased with increasing doses of IBG (Figure 5A). In addition, the effect of IBG on blood biochemical levels (ALP, AST, ALT, CREA, UREA) was assessed. The results showed that IBG significantly (*p* < 0.01) reduced the blood biochemical enzyme levels in the infected mice (Figure 5). Tissue sections showed that IBG significantly alleviated pathological damage in infected mice, including liver hemorrhaging and hepatocyte necrosis, renal tubular epithelial cell degeneration and necrosis, and lung hemorrhaging (Figure 6). In conclusion, IBG showed good therapeutic effects in SS3-infected mice.

## 3. Discussion

The development of new antibiotics lags far behind the development of drug resistance. Meanwhile, the exploration of new targets and new chemical structures of antibacterial drugs is an important direction for the development of antibacterial drugs [34]. Guanidine functional groups have an important place in medicinal chemistry because of their high stability and base modulation [35]. The high p*K*a value of the guanidine group, protonated under physiological conditions, is the main reason for its pharmacological action, while the versatility of its side chains confers a high degree of modifiability of its physical properties to adapt to different use scenarios. Chitosan derivatives containing a guanidine functional group, synthesized by a direct reaction between chitosan and cyanamide, have broad-spectrum antibacterial activity and can effectively inhibit a variety of bacteria, including *Escherichia coli*, *Staphylococcus aureus*, and *Pseudomonas aeruginosa* [36]. Biguanide-derived nanoparticles FTP NPs showed potent anti-MRSA activity, both inside and outside biofilms, with the bactericidal mechanism involving membrane permeabilization [37].

Similar to other guanidine derivatives, IBG can disrupt the integrity of *S. suis* cell membranes through electrostatic interactions with negatively charged cell envelopes. In previous studies, we also found that IBG had antibacterial activity against *Enterococcus* and *S. aureus*, with the best antibacterial activity against *S. aureus* (0.125–4 μg/mL) and the worst against *S. suis* (0.25–8 μg/mL), which may be related to the differences in bacterial membrane composition [27,28]. The structural and functional integrity of cell membranes is essential for cell survival, and therefore, cell membranes are worth considering as targets for antimicrobial drugs [38]. From a safety perspective, IBG is not genotoxic, reproductive genotoxic, embryotoxic, or teratogenic and has an oral acute toxicity classification of low toxicity in rats (LD_50_ of 1870.83 mg/kg b.w.) [39]. The main guanidine derivatives currently in clinical use are metformin and robenidine. The biguanides, metformin, phenformin, and buformin are a class of hypoglycemic drugs developed in the 1950s for the treatment of type 2 diabetes, and only metformin is now approved for use in most countries [40]. Phenformin and buformin have been withdrawn from the market due to an increased risk of lactic acidosis [41].

More important for *S. suis* is its virulence [42]. Of course, IBG is not only bactericidal against *S. suis*, but it also reduces the virulence of *S. suis* in vivo and in vitro, such as suilysin (Sly). Suilysin plays an important role in the inflammatory response by destroying eukaryotic cells and inducing the release of cytokines from immune cells [43]. Given that suilysin is a key virulence factor for *S. suis* for the colonization of host cells, escape from host immunity, and the production of cytotoxicity, suilysin is considered a new target to explore anti-virulence compounds for the treatment of *S. suis* infections [44,45]. In the present study, IBG showed excellent hemolytic activity against *S. suis* ATCC43765 culture supernatant and significantly inhibited the expression of the *Sly* gene. At the same time, IBG had no significant hemolytic activity in the therapeutic range. Compared with IBG, quercetin, piceatannol, and baicalein only inhibited the hemolytic activity of Sly protein and reduced the inflammatory response to streptococcus suis infection, but had no anti-streptococcal activity [46,47,48]. In conclusion, IBG has a more comprehensive function in clinical use.

In terms of chemical structure, IBG is a two-part symmetrical diaminoguanidine derivative containing two isopropoxyphenyls, which means that it is highly lipid-soluble and poorly water-soluble, severely limiting the route to clinical use. The most chemically similar compound to IBG is robenidine, which differs only in the substituent on the benzene ring [IBG: 4-OCH(CH_3_)_2_, robenidine: 4-Cl] (Figure 1A). Robenidine is widely used in the treatment of coccidial infections in poultry and rabbits, and its antimicrobial efficacy was later evaluated when it was found to be more effective against Gram-positive bacteria, *S. pseudintermedius* and *beta-hemolysis* streptococcs, and showed good synergistic effects when used in combination with EDTA against Gram-negative bacteria [49]. In the structure-activity relationship (SAR) investigation of robenidine, it was found that the antibacterial activity of the 4-OH robenidine analog was significantly reduced, and its methylated analog (4-OCH_3_) had no antibacterial activity, in contrast, whereas the alkyl substituent analog was more effective, with 4-CH_3_ and 4-CH(CH_3_)_2_ being more prominent [50,51]. This shows that 4-OCH(CH_3_)_2_ is essential for the antibacterial activity of IBG. Obviously, the search for a more optimal chemical structure is still a work in progress.

## 4. Materials and Methods

### 4.1. Bacterial Strains and Chemicals

The *S. suis* strain, ATCC 43765, and 12 clinical *S. suis* isolates were used in this study (Table 1). All strains were grown in tryptose soya broth (TSB; Hopebio, Qingdao, China) or plated on tryptose soya agar (TSA; Hopebio, Qingdao, China) with 5% (*v/v*) newborn fetal bovine serum at 37 °C. Isopropoxy benzene guanidine (IBG; purity: 99.9%) was synthesized by the Guangzhou Insighter Biotechnology Co., Ltd. (Guangzhou, China).

### 4.2. Antimicrobial Susceptibility Testing

The minimum inhibitory concentration (MIC) of isopropoxy benzene guanidine (IBG) was determined according to the Clinical and Laboratory Standards Institute (CLSI) guidelines. The microbroth dilution method was applied to 96-well plates with Mueller–Hinton broth (MHB; 5% newborn fetal bovine serum; Hopebio, Qingdao, China). The overnight bacteria cultures were adjusted to 5 × 10^5^ colony-forming units (CFU)/mL. After 18 h of incubation at 37 °C, the MICs were defined as the lowest antibiotic concentration without bacterial growth by visual inspection.

### 4.3. Kill Kinetic Assay

The time-dependent killing of *S. suis* ATCC 43765 with various concentrations of IBG was investigated. The overnight cultures of *S. suis* ATCC 43765 were diluted by 1:100 and treated with 0, 8, 16, 32, and 80 μg/mL of IBG. Colony counts were obtained by serial dilutions and plate counts at 0, 1, 2, 3, 4, 6, and 8 h. The time-kill curves were plotted as log CFU/mL versus time.

### 4.4. Resistance Studies

Cultures were serially passaged for 28 days while being treated with various concentrations of IBG in order to better understand how resistance to IBG develops. Overnight cultures of *S. suis* ATCC 43765 were inoculated into MHB containing 1×, 2×, 4×, and 10× MIC of IBG. The ciprofloxacin-treated group was used as a positive control. The MIC of each generation of cultures against IBG or ciprofloxacin was measured by the microbroth dilution method using 96-well plates. Fold changes in MICs for IBG and ciprofloxacin compared to initial values were calculated.

### 4.5. Membrane Integrity Test

To assess the effect of IBG on the integrity of bacterial membranes in *S. suis* ATCC 43765, the membrane permeability of *S. suis* ATCC 43765 induced by IBG was tested using the fluorescent dye propidium iodide (PI; Aladdin, Shanghai, China) at a final concentration of 10 nM. The *S. suis* ATCC 43765 cells were washed three times with PBS (pH = 7.4) and adjusted to a final concentration of OD_600_ nm of 0.5. After incubation with PI at 37 °C for 20 min, *S. suis* ATCC 43765 was treated with different final concentrations of IBG (8, 16, 32, and 80 μg/mL) for 1 h, and the fluorescence was measured at excitation wavelength 535 and emission wavelength 615.

### 4.6. Membrane Potential Assay

The fluorescent probe DiSC_3_(5) (Thermo Scientific, Massachusetts, United States) was used to determine the effect of IBG on the cell membrane potential (ΔΨm) of *S. suis.* In total, overnight cultures of *S. suis* ATCC 43765 were washed three times with HEPES containing 20 mM glucose, and the bacterial suspensions were adjusted to an OD_600_ nm of 0.5. DiSC_3_(5) was then added to a final concentration of 5 μM and incubated at 37 °C for 20 min to plateau. Cultures were treated with IBG at final concentrations of 8, 16, 32, and 80 μg/mL, and changes in DiSC_3_(5) fluorescence values were detected by spectrofluorometer at Ex.622 nm/Em.670 nm over 60 min.

### 4.7. ΔpH Assay

ΔpH, another component of the proton motive force (PMF), was measured by pH-sensitive fluorescent probes BCECF-AM. *S. suis* ATCC 43765 were washed and resuspended to obtain an OD_600_ nm of approximately 0.5 with 5 mM HEPES containing 20 mM glucose. After treatment with BCECF-AM for 20 min, various concentrations of IBG were added, and then ΔpH of *S. suis* ATCC 43765 was measured using the excitation wavelength of 488 nm and emission wavelength of 535 nm for 60 min.

### 4.8. ATP Determination

Intracellular and extracellular ATP levels of *S. suis* ATCC 43765 were detected by an Enhanced ATP Assay Kit (Beyotime, Shanghai, China). The overnight cultured *S. suis* ATCC 43765 cells were washed three times with PBS (pH = 7.4) and resuspended to an OD_600_ nm of 0.5. Different concentrations of IBG were added to the resuspension and incubated at 37 °C for 1 h. The bacterial cultures were then centrifuged at 12,000 rpm for 5 min at 4 °C. The supernatants were collected for the determination of extracellular ATP levels. Meanwhile, the precipitates were lysed with lysozyme, and the intracellular ATP was detected after centrifugation. The assay working solution was added to a 96-well plate and incubated for 5 min at room temperature. The supernatant was then added to a 96-well plate, rapidly mixed, and then ATP levels were measured using an EnSight^®^ Multimode Plate Reader.

### 4.9. ROS Detection

Reactive oxygen species (ROS) levels of IBG-treated *S. suis* ATCC 43765 were measured by ROS Assay Kit (Beyotime, Shanghai, China). Briefly, the overnight cultured *S. suis* ATCC 43765 cells were washed with PBS (pH = 7.4) and resuspended to an OD_600_ nm of 0.5. DCFH-DA (10 µM) was added to the culture and incubated at 37 °C for 20 min. Different concentrations of IBG were then added to treat the cultures for 60 min, and the fluorescence values were measured with 488 nm excitation and 525 nm emission filters.

### 4.10. Safety Assessment

The hemolytic activity of IBG was determined by defibrillating sheep blood erythrocytes. Briefly, 2% defibrillated sheep blood erythrocytes were treated with various concentrations of IBG (0–80 μg/mL) for 1 h. A PBS (pH = 7.4) treatment, with or without 2.5% Triton X-100, was used as positive and negative controls, respectively. The fluorescence value at OD_543_ nm was measured to evaluate the rate of hemolysis, and the following formula was used to evaluate the rate of hemolysis, Equation (1): (1)$$Haemolysise\;\%=\frac{OD_{sample}-OD_{\mathrm{negative}\;\mathrm{controls}}}{OD_{\mathrm{positive}\;\mathrm{controls}}-OD_{\mathrm{negative}\;\mathrm{controls}}}\times 100\;\%$$

### 4.11. Evaluation of the Effect of IBG on the Hemolysis Activity of the Culture Supernatant of S. suis ATCC 43765

The overnight cultured *S. suis* ATCC 43765 was centrifuged at 12,000 rpm for 10 min at 4 °C. The supernatants were collected and incubated with different concentrations of IBG at 37 °C for 30 min. After adding 2% defibrillated sheep blood cells to the culture for 1 h, the culture was centrifuged at 1000 rpm for 5 min at 4 °C, and the supernatant was collected to measure its absorbance at OD_543_ nm. Samples treated with 2.5% Triton X-100 were set as positive controls. The PBS (pH = 7.4) treatment served as a negative control.

### 4.12. RT-qPCR

RT-qPCR was used to determine the effect of IBG on *Sly* gene expression. The overnight cultured *S. suis* ATCC 43765 cells were transferred 1:100 to fresh TSB medium and grown at 37 °C with shaking at 200 rpm to the logarithmic growth phase. After the cultures were washed with PBS (pH = 7.4) and incubated with different final concentrations of IBG for 4 h, total RNA was extracted, and cDNA was synthesized using the PrimeScript™ 1st Strand cDNA Synthesis Kit (Takara, Japan) according to the manufacturer’s instructions. A real-time PCR assay was performed with the SYBR Premix Ex Taq (Takara, China) following the manufacturer’s instructions. The primers used are listed in Table 2 [52]. The 16S rRNA served as the endogenous control. Relative expression levels of the *Sly* gene were calculated by the 2^−ΔΔ*CT*^ method.

### 4.13. Molecular Docking

The model structure of the Sly protein was found in the UniProt Knowledgebase (https://www.uniprot.org/uniprotkb accessed on 25 October 2022). The protein sequence was A4FS11. The 2D structure of IBG was displayed using ChemDraw 20.0. The LibDock protocol of Discovery Studio 2019 was used to perform the molecular docking of the Sly protein and IBG.

### 4.14. Establishment of a Mouse Model of Intraperitoneal Infection

Male BALB/C mice, 6 weeks old, were obtained from the Guangdong Medical Laboratory Animal Center. The mice were randomly divided into five groups of six mice each. All procedures used in this experiment were performed according to the guidelines developed by the Laboratory Animal Center of South China Agricultural University. After 5 days of acclimatization, mice were injected intraperitoneally with 100 μL of a 1.5 × 10^8^ CFU culture of *S. suis* SS3. One hour later, infected mice were treated with PBS and IBG (8, 16, 32, 80 mg/kg) by intraperitoneal injection. After 7 days, the survival curve of the mice was constructed based on the detection data.

In addition, the effect of IBG on blood biochemical enzymes in *S. suis* SS3-infected mice was evaluated, and the mice were treated as described above. Mice (six per group) were injected intraperitoneally with 100 µL of 1.5 × 10^8^ CFU of *S. suis* SS3 culture, and 1 h later with PBS and different doses of IBG (8, 32 mg/kg) were injected as described above. After 12 h of treatment, cardiac blood was collected from anesthetized mice and assayed for the levels of alanine transaminase (ALT), aspartate transaminase (AST), creatinine (CREA), UREA, and alkaline phosphatase (ALP). Meanwhile, the lung, kidney, spleen, and liver tissues were collected, ground, diluted, and counted on a drop plate to determine the amounts of infected bacteria. Finally, the lung, kidney, spleen, and brain tissues of the mice were fixed with 4% paraformaldehyde, and their pathological changes were analyzed.

### 4.15. Statistical Analysis

GraphPad Prism 9.0 software was used for statistical analysis. All data were expressed as mean ± standard deviation. ns, not significant. * *p* < 0.05, ** *p* < 0.01, *** *p* < 0.001, **** *p* < 0.0001.

## 5. Conclusions

IBG shows surprising anti-streptococcal activity in vivo and in vitro. IBG acts by disrupting the physical structure and function of bacterial cell membranes. It is a concentration-dependent, fast-acting bactericidal agent and is not susceptible to induced resistance. In addition, suilysin (sly) is one of the major virulence factors of *S. suis* and is closely associated with pathogenicity. IBG shows an exceptional ability to reduce the hemolysis activity of *S. suis* by reducing the expression of the *sly* gene and binding to suilysin. Taken together, it is clear that IBG is an excellent potential antimicrobial agent for the treatment of *S. suis* infections.

## Figures and Tables

**Figure 1 ijms-24-07354-f001:**
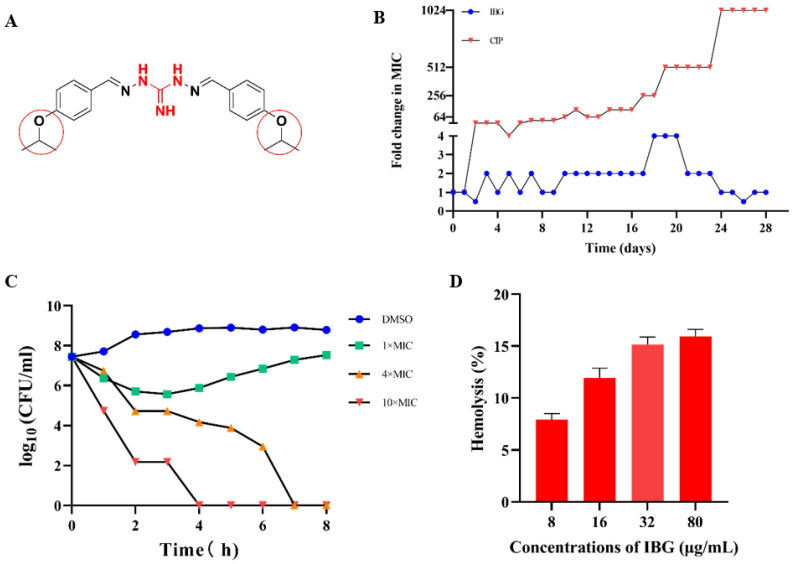
IBG is a potential antimicrobial agent against *S. suis*. (**A**) The chemical structure of IBG, the guanidine group, is labeled in red, and 4-OCH(CH_3_)_2_ is marked with a red circle. (**B**) No resistance to IBG occurred in the 28-day serial passage. CIP, ciprofloxacin, was used as a control group. (**C**) Time-kill curves of IBG against Streptococcus suis with 1 × MIC (8 μg/mL), 4 × MIC (32 μg/mL), and 10 × MIC (80 μg/mL) concentrations. The 1% DMSO-treated group served as the control group. (**D**) Hemolytic activity of IBG on defibrillated sheep blood erythrocytes.

**Figure 2 ijms-24-07354-f002:**
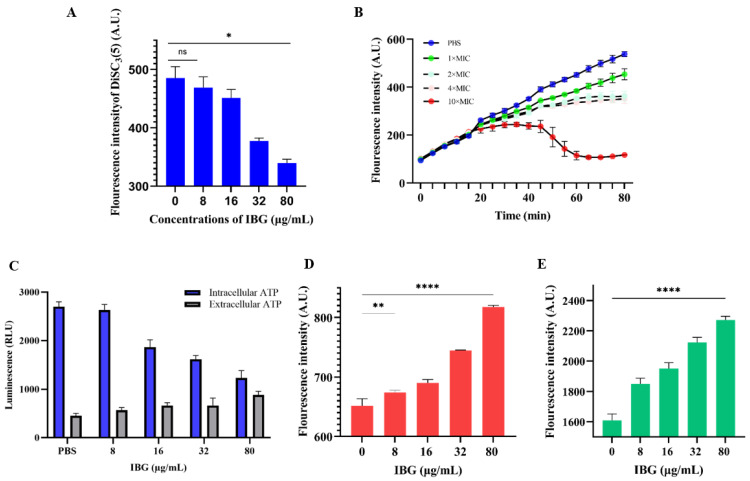
The antibacterial mechanism of IBG against Streptococcus suis. (**A**) Fluorescent probe DiSC_3_(5) was used to detect the membrane potential (ΔΨm). Disruption of membrane potential in IBG-treated *S. suis* ATCC 43765. *p* values were determined using a non-parametric one-way ANOVA. (**B**) The ΔpH of *S. suis* ATCC 43765 treated with different concentrations of IBG was detected by the fluorescent probe BCECF-AM. (**C**) The intracellular and extracellular ATP levels of IBG-treated *S. suis* ATCC 43765. (**D**) The effect of low concentrations of IBG on ROS accumulation in *S. suis* ATCC 43765 was negligible. (**E**) The permeability of *S. suis* ATCC 43765 membranes was probed by PI. 1 × MIC, 8 μg/mL; 2 × MIC, 16 μg/mL; 4 × MIC, 32 μg/mL; and 10 × MIC, 80 μg/mL. *, *p* < 0.01, ****, *p* < 0.0001.

**Figure 3 ijms-24-07354-f003:**
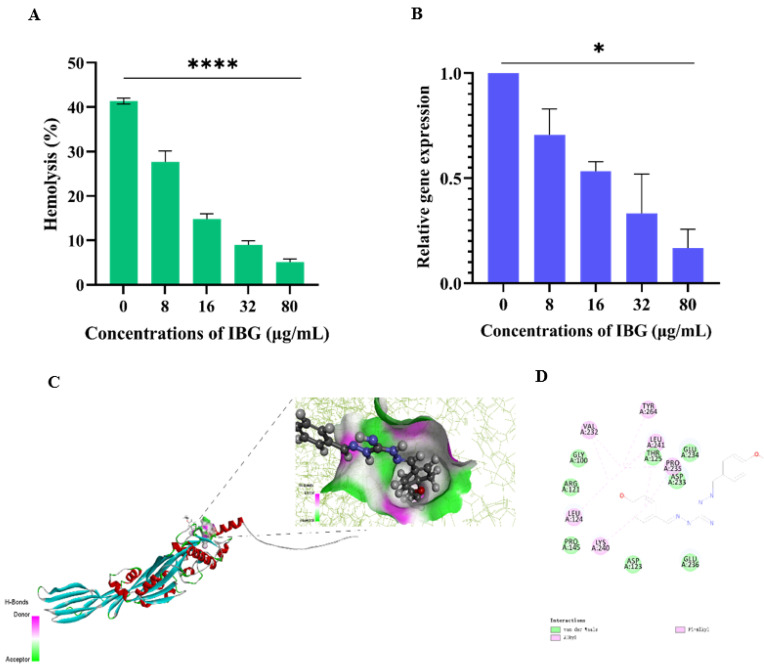
IBG alleviated the hemolytic activity of *S. suis* ATCC 43765. (**A**) Hemolytic activity of supernatants of *S. suis* ATCC 43765 cultures treated with different concentrations of IBG. (**B**) Relative expression levels of the Sly gene of *S. suis* ATCC 43765 in the presence of IBG. (**C**,**D**) The interaction pattern of IBG and Sly proteins. *, *p* < 0.01, ****, *p* < 0.0001.

**Figure 4 ijms-24-07354-f004:**
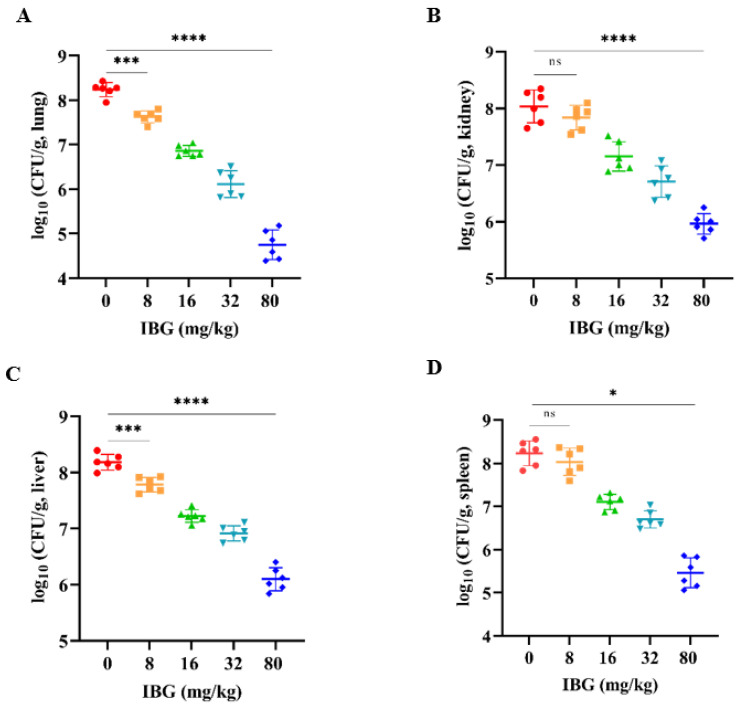
Bacterial load of (**A**) lung, (**B**) kidney, (**C**) liver, and (**D**) spleen tissues of SS3-infected mice after treatment with different doses of IBG. IBG significantly reduced the bacterial load of lung, kidney, liver, and spleen tissues. An unpaired two-tailed *t*-test was used for statistical analysis. *, *p* < 0.01, ***, *p* < 0.001, ****, *p* < 0.0001.

**Figure 5 ijms-24-07354-f005:**
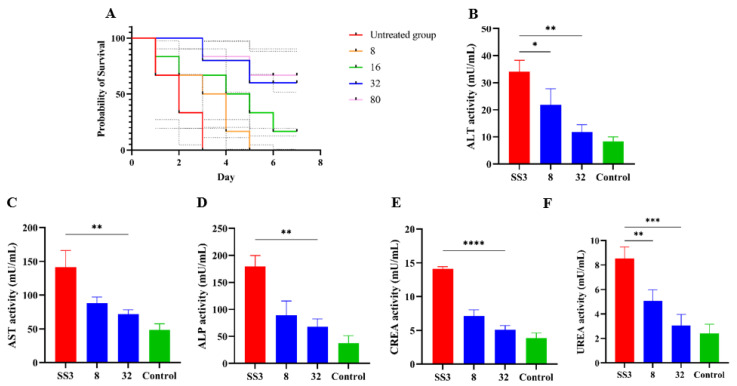
The efficacy of IBG against *S. suis* in vivo. (**A**) The survival curve of various concentrations of IBG cured *S. suis* SS3-infected mice model. Blood levels of (**B**) ALT, (**C**) AST, (**D**) ALP, (**E**) CREA, and (**F**) UREA in mice. SS3 is the infected group; eight were in the treated group with 8 μg/mL IBG, and 16 were in the treated group with 16 μg/mL IBG; Control is the blank group without any treatment. *, *p* < 0.01, **, *p* < 0.01, ***, *p* < 0.001, ****, *p* < 0.0001.

**Figure 6 ijms-24-07354-f006:**
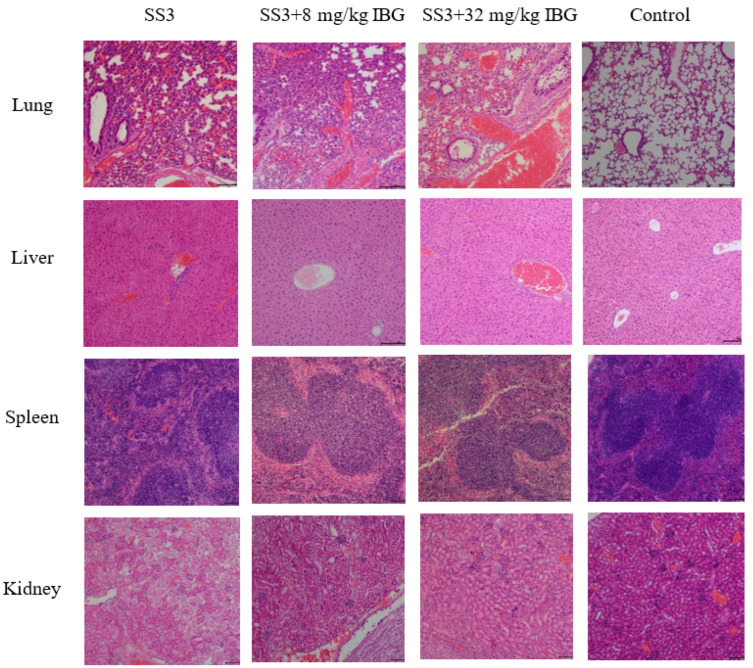
Histologic analysis of different organs using hematoxylin-eosin (HE) staining. Pathological changes in lung, liver, spleen, and kidney histology after IBG treatment. IBG alleviated the pathological damage and inflammatory response in infected mice.

**Table 1 ijms-24-07354-t001:** Activity of IBG against a wide variety of MLST *S. suis*. MLST, MLST, multi-locus sequence type; MIC, minimum inhibitory concentration; /, none.

Isolate	MLST	MIC (mg/L)	ARGs
*S. suis* ATCC 43765	/	8	/
SS3	7	8	*msr*(D), *tet*(O)
SS12	850	8	*ant(6)-la*, *aac(6′)-aph(2″)*, *erm*(B), *lsa€*
SS14	27	8	/
SS16	243	8	*tet*(O)
SS23	1	8	*tet*(O)
SS24	242	8	*erm*(B), *tet*(40), *tet*(O)
SS25	308	8	*ant(6)-la*, *tet*(32)
SS26	839	8	*tet*(O)
SS28	25	8	*erm*(B), *tet*(O)
SS32	28	8	*erm*(B), *tet*(O)
SS36	94	8	*tet*(O)
SS40	87	8	*erm*(B), *tet*(O)

**Table 2 ijms-24-07354-t002:** The primers used for RT-qPCR in this study.

Primer	Forward Primer	Reverse Primer
*16S rRNA*	GTTGCGAACGGGTGAGTAA	TCTCAGGTCGGCTATGTATCG
*Sly*	TCATTCAGGTGCTTATGTTGCG	GAAGATTGCGAGCATTTCCTGG

## Data Availability

The data presented in this study are available in article.

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
