# Peer review of "Isopropoxy Benzene Guanidine Ameliorates *Streptococcus suis* Infection In Vivo and In Vitro"

_ijms, 2023, doi:10.3390/ijms24087354_

Round 1
Reviewer 1 Report
Reviewer report about manuscript by N. Han and co-authors entitled ‘Isopropoxy benzene guanidine Ameliorates Streptococcus suis Infection in Vivo and in Vitro’ submitted for publication to International journal of Molecular Science
The evaluated manuscript is focused on a highly actual and complex medical problem of discovery of novel antimicrobial drugs namely intended for control of Streptococcus suis, of a highly dangerous animal pathogen. The work is carried out with involvement of a number of modern efficient methods and is interesting for a broad auditory. Despite the large amount of reliable data, the manuscript requires a major revision for the following reasons:
1. The introduction section does not provide sufficient knowledge about state-of the art in the specific area of the study. As far, as S. suis was chosen as the only specific target of IBG, the intraspecies polymorphism of pathogenicity determinants should be described here. Table 1 contains a column ‘Mult-locus type. Nomenclature of this genetic determinant must be described in the Introduction. Known serological types of S. suis and their pathogenic impact should be discussed. The history of discovery of IBG and principal known data about its antimicrobial activity should be discussed more extensively. Its activity towards other microorganisms beyond S. suis should be characterized. Origination of IBG preparation and its purity should be described here or in Material and methods section. The items of systemic toxicity of IBG and other substituted guanidinia should be discussed. List of the cited publication should be respectively extended.
2. Treatment duration of bacterial with IBG is not specified almost in all sub-sections of Materials and methods.
3. The authors should include data about S. suis ATCC 43765 trails to Table 1 and discuss reasons of apparently different susceptibility of wild-type and collection strains to IBG. Following data of Fig 1C, S. suis ATCC 43765 is not sensitive to the tested drug in the concentration 8 µg/ml, whereas all tested wild-type strains of S. suis are completely killed by this concentration of the drug.
4. The authors should explain why the used a wild-type strain of S. suis for in vivo biological trails whereas S. suis ATCC 43765 was used in all in vitro experiments.
5. The term ‘hemolysis’ should not be applied towards lysis of bacterial cells.
6. Cultivation of S. suis should be described in more detail. The described procedure of cfu counting after treatment with IBG looks irreproducible. The authors should specify the acceptable period of the culture storage withing those the cfu count can be supposed unchanged and give an experimental confirmation of this.
Taking into account the listed criticisms, I recommend the authors to carry out a major revision of their text.

Reviewer 2 Report
The authors have studied isopropoxy benzene guanidine (IBG) significantly attenuated the effects caused by S. suis infection in vivo and in vitro by killing S. suis and reducing S. suis pathogenicity. Additionally, the experiments demonstrated that IBG disrupted the integrity of S. suis cell membranes and increased the permeability of S. suis cell membranes
Overall, the study showed IBG can be a promising compound to treat S. suis infections.
Comments:
Figure 4. Needs detailed captions. The captions are too brief and end abruptly.
Figure 2. Captions are missing.
Table 2. could be moved to supplementary section.
The English structure and grammar needs attention.
Please provide conclusions section with important findings and key implications of the study.
Statistical significant data needs to be emphasized in abstract and conclusions.
Reviewer 3 Report
Rise of antibiotic resistance bacteria has been led to life threatening infections. There is unmet need to identify new antibacterial agents. Authors in the current study identified new antibacterial agent ‘isopropxy benzene guanidine’ and showed its antibacterial activity on Streptococcus suis which is zoonotic pathogen causes serious infections in both animals and humans.
overall, authors experimental approaches both invitro and invivo in determining the efficacy of isopropxy benzene guanidine seems good.
Manuscript writing and demonstration of results with relevant figures are good.
Round 2
Reviewer 1 Report
The revisions done by the authors provide sufficient amendment adequate to the former criticisms. I would like to recommend spell-checking taking into account an erroneous spelling of Streptococc (not Streptococcus) in line 334. The references (numbers in brackets) should be placed within respective sentences where they are quoted, not after the final dot